# Comprehensive Evaluation of Lipid Nanoparticles and Polyplex Nanomicelles for Muscle-Targeted mRNA Delivery

**DOI:** 10.3390/pharmaceutics15092291

**Published:** 2023-09-07

**Authors:** Xuan Du, Erica Yada, Yuki Terai, Takuya Takahashi, Hideyuki Nakanishi, Hiroki Tanaka, Hidetaka Akita, Keiji Itaka

**Affiliations:** 1Department of Biofunction Research, Institute of Biomaterials and Bioengineering, Tokyo Medical and Dental University (TMDU), Tokyo 101-0062, Japan; xuandu.bif@tmd.ac.jp (X.D.); erika.bif@tmd.ac.jp (E.Y.); terai.bif@tmd.ac.jp (Y.T.); t-takahashi.orth@tmd.ac.jp (T.T.); h.nakanishi.bif@tmd.ac.jp (H.N.); 2NANO MRNA, Co., Ltd. Tokyo 104-0031, Japan; 3Graduate School of Pharmaceutical Sciences, Tohoku University, Sendai 980-8578, Japan; hiroki.tanaka.e1@tohoku.ac.jp (H.T.); hidetaka.akita.a4@tohoku.ac.jp (H.A.); 4Clinical Biotechnology Team, Center for Infectious Disease Education and Research (CiDER), Osaka University, Osaka 565-0871, Japan

**Keywords:** messenger RNA (mRNA), lipid nanoparticle (LNP), polyplex nanomicelle

## Abstract

The growing significance of messenger RNA (mRNA) therapeutics in diverse medical applications, such as cancer, infectious diseases, and genetic disorders, highlighted the need for efficient and safe delivery systems. Lipid nanoparticles (LNPs) have shown great promise for mRNA delivery, but challenges such as toxicity and immunogenicity still remain to be addressed. In this study, we aimed to compare the performance of polyplex nanomicelles, our original cationic polymer-based carrier, and LNPs in various aspects, including delivery efficiency, organ toxicity, muscle damage, immune reaction, and pain. Our results showed that nanomicelles (PEG-PAsp(DET)) and LNPs (SM-102) exhibited distinct characteristics, with the former demonstrating relatively sustained protein production and reduced inflammation, making them suitable for therapeutic purposes. On the other hand, LNPs displayed desirable properties for vaccines, such as rapid mRNA expression and potent immune response. Taken together, these results suggest the different potentials of nanomicelles and LNPs, supporting further optimization of mRNA delivery systems tailored for specific purposes.

## 1. Introduction

Messenger RNA (mRNA) has been well investigated for potential clinical applications, including infectious diseases, cancers, and genetic disorders [1,2,3]. Over the past decade, mRNA-based therapy has been increasingly favored over conventional gene therapy or protein-replacement therapy because of its versatility, safety, and the relatively lower production cost [3]. In particular, the recent success of two mRNA vaccines against COVID-19 produced by Moderna and Pfizer-BioNTech heralds a paradigm shift in biomedical research. To overcome hurdles such as short half-life, rapid degradation by endosomal enzymes, and recognition by Toll-like receptors (TLRs) upon cell entry, ongoing efforts, including codon optimization, nucleoside modification, and cap modification, have been made to improve the translation and stability of mRNA [1,2,4].

Although nuclear entry is not required for mRNA therapy to function, cell entry is still a challenge, since highly negatively charged mRNA cannot be efficiently internalized into the lipid bilayer of the cell membrane [2]. To effectively deliver mRNA into the cytosol, various platforms, such as lipid nanoparticles (LNPs) [5], have been employed to facilitate in vivo mRNA delivery. Typically composed of ionizable lipids, helper lipids, cholesterol, and polyethylene glycol (PEG) lipids, LNPs have shown great promise for the delivery of mRNA-based vaccines. Apart from COVID-19, LNPs have also been used in mRNA vaccines against several diseases, including rabies, influenza, and melanoma [6,7,8]. However, despite being commonly used for in vivo mRNA delivery in preclinical studies, issues such as immunogenicity and toxicity predominantly associated with the ionizable lipids remain to be addressed [9,10,11], which may limit LNPs’ potential for therapeutic applications.

Based on this context, we intended to compare the performance of polyplex nanomicelles, our original cationic polymer-based mRNA carrier, with that of LNPs under the same conditions. Formed by the electrostatic interaction between mRNA and block-copolymer PEG-PAsp (DET), polyplex nanomicelles have proven effective and well-tolerated in a wide range of disease models, such as osteoarthritis (OA) [12], temporomandibular joint osteoarthritis (TMJ-OA) [13], spinal cord injury [14], and autism-like behaviors [15]. In this study, we aimed to evaluate the two mRNA carriers in various aspects, including delivery efficiency, organ toxicity, muscle damage, immune response, and pain. By systematically comparing and evaluating diverse mRNA delivery platforms, we hope to gain a deeper understanding of their strengths, limitations, and suitable applications. Such information will be essential for the development of more efficient and safer delivery systems, ultimately facilitating the clinical translation of mRNA-based therapies. Nevertheless, to our knowledge, there have been few reports on this topic in the existing literature. Therefore, this study holds importance in addressing this crucial knowledge gap and contributing to the advancement of mRNA delivery systems.

The skeletal muscle has been extensively studied as the target tissue for mRNA delivery due to its large size and efficient vascularization. Moreover, the abundance of immune cells in the muscle makes it the optimal site for vaccination, which further accerlates the development of mRNA vaccines. Despite the wide application of intramuscular (IM) injection for mRNA-based vaccines in both preclinical and clinical studies [4], problems such as limited injection volume, uneven distribution, and damage to surrounding tissues remain to be solved [16]. In this regard, the hydrodynamic limb vein (HLV) injection, also referred to as limb perfusion, has been proposed as a promising route for nonviral gene delivery to the skeletal muscle. The HLV approach involves the isolation of the limb and injection of a large volume of nucleic acids, resulting in a transient increase in blood pressure and facilitating distribution throughout the target tissue [16,17,18]. Compared to IM injection, HLV delivery could achieve homogeneous and widespread drug distribution, which is more favorable considering therapeutic needs. Indeed, the HLV method has demonstrated successful results in a variety of animal models [19,20,21,22,23,24,25], and was well-tolerated in patients with muscular dystrophy [26,27]. Given that both the delivery system and the administration route could impact the mRNA expression kinetics and biodistribution [28,29,30], we also sought to compare the performance of nanomicelles and LNPs delivered via IM or HLV approaches.

## 2. Materials and Methods

### 2.1. Preparation of mRNA-Loaded LNP and Polyplex Nanomicelle

CleanCap Firefly Luciferase (FLuc) mRNA (5-methoxyuridine) was purchased from TriLink Biotechnologies, San Diego, CA, USA (L-7202).

LNPs were formulated according to a previous report [31]. Ionizable lipid SM-102 was purchased from Cayman Chemical, MI, USA. Helper lipid DSPC (1,2-distearoyl-sn-glycero-3-phosphocholine), cholesterol, and PEG lipids (DMG-PEG-2000) were purchased from NOF CORPORATION, Tokyo, Japan. Briefly, these lipids were dissolved in ethanol, and a lipid mixture with a molar ratio of 50:10:38.5:1.5 (SM-102:DSPC:cholesterol:PEG) was mixed with 50 mM sodium citrate buffer (pH 4.0) containing mRNA in NanoAssemblr Ignite at a flow ratio of 3:1 (buffer:ethanol) and a total flow rate of 2 mL/min. The N:P ratio (positively charged amine groups to negatively charged phosphates) was 5.67:1. The external buffer was replaced with Tris Buffered Saline (147 mM NaCl, 2.7 mM KCl, and 100 mM Tris) using Amicon Ultra Centrifugal Filter Units. The diameter of LNP particles was determined by dynamic light scattering to be between 50 and 120 nm, with an average size of 84.87 nm. The Fluc mRNA-loaded LNP solution was kept at 4 °C and the experiments were performed within two weeks after preparation.

For preparing mRNA-loaded nanomicelle, a block copolymer of PEG-PAsp(DET) was synthesized according to a previous report [32]. The molecular weight of PEG was 43,000 g/mol, and the polymerization degree of PAsp(DET) was determined to be 63 by 1H NMR analysis. Briefly, the solutions of PEG-PAsp(DET) and FLuc mRNA dissolved in 10 mM HEPES buffer (Wako, Japan) were mixed thoroughly to form the nanomicelle. The N:P ratio (positively charged amines in the polymers to negatively charged phosphates in mRNA) was fixed to 8:1, and the mRNA concentration was set to 200 μg/mL in the final solution. The size of polyplex nanomicelle was found to be around 50 nm in diameter, as previously reported [33]. The mRNA-containing nanomicelle was freshly prepared before each experiment.

The particle size, polydispersity (PDI), and zeta potential of LNPs and nanomicelles, measured by dynamic light scattering measurements are listed in Appendix A.

### 2.2. Animal Experiments

All animal experimental procedures were approved by the Animal Committee of Tokyo Medical and Dental University. Eight-week-old male C57BL/6J mice weighing approximately 20 g to 25 g (Sankyo Labo, Tokyo, Japan) were maintained under a 12 h light/dark cycle with ad libitum access to food and drinking water at Tokyo Medical and Dental University (TMDU). Mice were habituated for one week prior to all experiments. Before injection, hair was shaved to expose the lower hindlimb. For intramuscular administration, 50 μL of mRNA-loaded LNP or nanomicelle solution containing 10 μg FLuc mRNA was injected into the gastrocnemius muscle. Hydrodynamic limb vein injection was performed as previously described [17]. Briefly, under 2% isoflurane inhalation, a tourniquet was applied to a proximal part of the hindlimb to temporarily restrict the blood flow. Fifty μL of mRNA-loaded LNP or nanomicelle solution was injected at the distal site of the great saphenous vein. The tourniquet was removed 5 min after the injection. All injections used 30G needles.

### 2.3. In Vivo Imaging

Luciferase expression was assessed using IVIS Lumina II (Xenogen, Almeda, CA, USA) at 8, 24, 48, 72, 168, and 336 h after FLuc mRNA injection. Hair was removed from both legs before imaging to increase the detection sensitivity. Mice were anesthetized with 2% isoflurane and injected intraperitoneally with 3 mg D-luciferin (Wako, Osaka, Japan). To ensure optimal distribution, the images were acquired ten minutes after D-luciferin injection with an exposure time of 60 s. The background value was determined by the average photon flux in the noninjected hindlimb of each mouse.

### 2.4. Tissue Luciferase Assay

Mice were euthanized by cervical dislocation. For organ samples, the mice were sacrificed 24 h after LNP or nanomicelle administration. For muscle samples, the gastrocnemius muscles were collected at the end of the in vivo imaging experiment. The gastrocnemius, liver, and both kidneys were rapidly isolated and snap-frozen in liquid nitrogen. Tissue samples were added with the Passive Lysis Buffer (Promega, Madison, WI, USA) and homogenized in a Multibeads Shocker (Yasuikikai, Osaka, Japan). To obtain tissue lysate, the homogenate was centrifuged at 12,000 rpm for 5 min at 4 °C, and the supernatant was collected. Luciferase activity was evaluated using the Luciferase Assay System (Promega) and a GloMax Navigator Microplate Luminometer (Promega) by mixing 100 μL assay buffer with 10 μL lysate in a 96-well plate. The plate was then immediately transferred to the luminometer for detection, with an exposure time of 1 s.

### 2.5. Blood Biochemistry

Blood was collected into a heparinized tube by cardiac puncture under 4% isoflurane inhalation. After centrifugation at 1000× *g* for 5 min at room temperature, the plasma was carefully transferred to a new 1.5 mL tube and stored at −80 °C until further processing. Plasma samples were tested with a blood chemistry analyzer (DRI-CHEM NX600, FUJIFILM, Tokyo, Japan) using the following slides: BUN-PIII, TBIL-PIII, GOT/AST-PIII, GPT/ALT-PIII, CPK-PIII, LDH-P (IFCC). The reference values were obtained from the animal supplier (Japan SLC Inc., Shizuoka, Japan) as well as previous reports [34,35,36].

### 2.6. Histology and Image Analysis

For tissue histology, muscles were snap-frozen in isopentane immersed in liquid nitrogen for 60 s, and 6 mm thick transverse cross-sections were obtained using a cryostat (CM3050S, Leica Biosystems, Wetzlar, Germany). For Evans Blue staining, mice were injected intraperitoneally with 200 μL of 10% Evans Blue (Wako) dissolved in PBS. A few hours later, the permeation of Evans Blue was confirmed by the color change in the ears and paws of the mice. ImageJ (version 1.53k) was used to analyze the histological images.

### 2.7. Quantitative Real-Time Polymerase Chain Reaction (qRT-PCR)

Total RNA was extracted from the muscle samples using the RNeasy Fibrous Tissue Mini Kit (Qiagen, Hilden, Germany). After reverse transcription using ReverTra Ace RT Master Mix with gDNA remover (TOYOBO, Osaka, Japan), the cDNA was then mixed with the PowerTrack SYBR Green Master Mix (Applied Biosystems, Waltham, CA, USA) and the gene-specific primers. qPCR was performed on a StepOnePlus Real-time PCR system (Applied Biosystems). The sequences of the primers used in this study are listed in Table 1. The expression level of the housekeeping gene *Actb* was used for normalization.

### 2.8. Static Weight-Bearing Testing

The incapacitance (static weight-bearing) test was performed to evaluate pain levels. The weight distribution on the injected hindlimb of the mice was assessed using an incapacitance tester (BIO-SWB-TOUCH-M, Bioseb, Vitrolles, France). This test measures the difference in weight distribution between the injected and noninjected hindlimb, and a distinct difference can be interpreted as the animal’s adaptation to pain [37]. Briefly, each mouse was placed in a chamber with its paws resting comfortably on two separate sensors, and the force (g) applied to the sensors was recorded over a 10 s period. The percentage of the total weight distributed on the injected hindlimb was determined as the static weight distribution. Mice were acclimated to the test chamber prior to regular testing.

### 2.9. Statistical Analysis

All data are presented as mean ± standard error of the mean (SEM). For statistical tests, one-way ANOVA followed by Tukey’s honest significance test or two-tailed Student’s *t*-test was used to assess the significance. *** *p* < 0.001, ** *p* < 0.01, * *p* < 0.05. Data analysis and visualization were performed using R (version 4.2.0) and Rstudio (version 2022.12.0+353).

## 3. Results

### 3.1. Evaluation of Luciferase Expression by In Vivo Imaging

Initially, we evaluated the delivery efficiency of LNP and polyplex nanomicelle using modified firefly luciferase (FLuc) mRNA (modified with 5-methoxyuridine) and in vivo imaging. The structures of FLuc mRNA-loaded nanomicelle and LNP were briefly illustrated in Figure 1A,B. Mice were given 50 μL intramuscular (IM) or hydrodynamic limb vein (HLV) injections containing 10 μg mRNA encapsulated in either LNP or polyplex nanomicelle, as illustrated (Figure 1C). LNP administered by the HLV technique displayed significantly higher expression at 8 h, which rapidly diminished within two days (Figure 2A). Conversely, FLuc delivered by nanomicelle via HLV injection had lower initial expression but peaked at 48 h, and remained steadily detectable for up to two weeks (Figure 2A,B). Intriguingly, the luciferase expression of LNP or nanomicelle by IM injection was lower than that by the HLV method. This is likely because HLV injection could target multiple muscle groups and enhance the overall expression [18].

In addition, since the signals were close to background levels at the end of the in vivo imaging experiment, we collected the muscle samples 2 weeks after the injection, and examined the FLuc expression by luciferase assay. Correspondingly, higher expression was observed in the group treated with nanomicelle via HLV administration (Figure 2C). These data imply that although LNP resulted in significantly higher expression initially, polyplex nanomicelle led to relatively sustained expression after administration.

### 3.2. Evaluation of Liver and Kidney Toxicity

Several studies have reported that LNPs could elicit systemic effects such as accumulation in the liver, even when administered intramuscularly [28,29,30,31]. We had similar observations by in vivo imaging, in which both HLV and IM administration of LNP resulted in immediate and abundant luciferase signals in the liver up to 24 h (Figure 2A). In contrast, no signals were detected in the nanomicelle group. Based on these findings, we conducted luciferase assays on the liver and kidney that were collected 24 h after the injection, since the liver and the kidneys are the major organs of drug metabolism and excretion. Luciferase expression in the liver after both IM and HLV LNP injections was almost 10,000-fold higher than that of the noninjected controls (Figure 3A), confirming off-target mRNA expression in the liver cells delivered by LNP. Additionally, luciferase expression was also found in the kidney of mice receiving LNP, although the signals were much milder in the HLV group than the IM group and were not statistically significant (Figure 3B). Meanwhile, no luciferase expression was detected in the liver or kidney of mice injected with nanomicelle.

To further determine whether polyplex nanomicelle or LNP could cause hepatic and renal toxicity, we analyzed blood biochemical markers associated with liver and kidney injury [38,39], since previous studies have reported an increase in liver enzymes such as the alanine aminotransferase (ALT) and aspartate aminotransferase (AST) in the blood following administration of LNP loaded with mRNA, or empty LNP in rodents [10,11]. We observed AST levels increased above normal (as indicated by the grey bands) on day one until day five in the groups of LNP-HLV and LNP-IM. Similarly, ALT levels showed an increase on day one in the LNP groups, but returned to normal level on day five. For the groups of nanomicelle, AST levels increased slightly above normal, but generally lower than those of the LNP groups. The levels of total bilirubin (TBIL) and blood urea nitrogen (BUN) were not remarkably changed from those of no injection control (Figure 3C). Collectively, these results indicate that LNP might induce slight damage on the liver cells, especially by IM injection, but the effect was transient. The toxicity of nanomicelle in the liver was present but minuscule. No toxicity was observed on the kidney.

### 3.3. Evaluation of Tissue Damage after Administration

Next, we inspected the muscle histology after administration of LNP or nanomicelle. An earlier study aimed at optimizing the LNP formulation for intramuscular delivery reported that some ionizable lipids could induce myofiber necrosis in a dose-dependent manner [31]. Two days after IM administration, mice injected with LNP showed widespread interstitial swelling and inflammatory cell infiltration, which appeared to become a mixed legion of necrosis, inflammation, and fiber degeneration on day 5 (Figure 4A,B).

Such histological changes were also reflected in the blood creatine kinase (CK) levels, although the differences in lactate dehydrogenase (LDH) were not significant (Figure 4C). In contrast, only a small area of myofibers along the injection tract underwent necrosis in mice treated with IM nanomicelle injection and was quickly recovered by day five (Figure 4B). Moreover, the muscle histology of mice injected with LNP or nanomicelle via HLV administration resembled that of the PBS group, with only sporadic inflammatory cells (Figure 4A,B). Interestingly, we noted a decrease in the average fiber area of mice five days after IM injection compared to mice receiving the same treatment by HLV (Figure 5A). This observation is likely due to satellite cell (muscle stem cell) activation, as indicated by small and centrally nucleated fibers [40]. No changes were found in the body weight (Appendix A), muscle mass, or the total myofiber area among all groups (Figure 5B,C). Taken together, these data demonstrate that the nanomicelle induced less myofiber damage than LNP.

To better understand the extent of muscle damage, we employed the Evans blue dye (EBD) staining method, which is a sensitive marker in the early stages of muscle damage [41]. Although blue in color when administered in vivo (Figure 6A), the dye exhibits red fluorescence (excitation at 620 nm, emission at 680 nm) [42], allowing examination at the microscopic level. Two days after IM injection, EBD signals were found in mice receiving IM injection of both nanomicelle and LNP, though it appeared that the distribution of EBD in the former group was confined to the fibers around the injected area and was more diffuse in the interstitial area in the latter. We found no EBD uptake in mice injected with mRNA encapsulated in either LNP or nanomicelle via the HLV route. This is further supported by the absence of EBD signals in PBS-treated mice by either route of administration, ruling out the possibility of muscle injury caused by the injection alone (Figure 6A,B). These data suggest that the HLV injection was less invasive than IM.

### 3.4. Evaluation of Inflammatory Response

Given the histological changes in the muscle, we examined the expression levels of inflammatory cytokines after administration of LNP or nanomicelle. In line with some previous reports of the inflammatory nature of the ionizable lipids used to formulate LNPs [9,10], we found a drastic increase in the muscle mRNA expression levels of interleukin-6 (IL-6) and interleukin-1β (IL-1β) 24 h following the IM injection of LNP (Figure 7A,B). Specifically, the increase in IL-1β expression was consistent with a recent study that identified the IL-1 and interleukin-1 receptor antagonist (IL-1ra) as the key regulators of the inflammatory response to the LNP/mRNA platform, which then triggers the release of proinflammatory cytokines including IL-6 [43]. HLV injection of LNP also induced a slight increase in the expression levels of IL-6 and IL-1β, but to a lesser extent compared with IM injection, although there were no significant differences between the two groups (Figure 7A,B). This discrepancy could be attributed to the fact that IM injection could rapidly activate the resident antigen-presenting cells (APC) in the muscle and the downstream proinflammatory pathways [5,44], whereas the HLV approach allows for a more homogeneous distribution of the injected fluid [16]. The TNF-α expression was also elevated in the LNP groups, although it was not statistically significant compared with no injection controls (Figure 7C). Conversely, the expression levels of the inflammatory cytokines were minimal in the groups receiving nanomicelle. These results further confirmed that the nanomicelle behaved in a less inducible manner of inflammation compared with LNP.

### 3.5. Evaluation of the Pain Level

Finally, we investigated the muscle pain after the injection of LNP or nanomicelle. Although pain at the injection site was the most commonly reported side effect by participants in the clinical trials of LNP/mRNA vaccines [45,46,47], little emphasis has been placed on the level of pain induced by LNP in preclinical animal studies. We sought to address this issue using the incapacitance test (static weight-bearing test), a well-established behavioral assay used to assess leg pain [37,48,49]. In addition to significant swelling and redness in the injected area, pain was observed in mice injected with intramuscular LNP at 24 h, which peaked at 48 h and slowly resolved within one week (Figure 8). The pain pattern looks similar to that of the participants in the clinical trials of mRNA vaccines [45,46,47]. In contrast, mice receiving nanomicelle or no injection showed almost no pain behaviors. These results further support the less invasive manner of the nanomicelle compared with LNP.

## 4. Discussion

Advances in materials science have led to a variety of mRNA delivery systems, but few studies have directly compared different mRNA delivery platforms under the same conditions. Here, we addressed this blind spot by thoroughly comparing LNPs with our original mRNA carrier polyplex nanomicelles in terms of delivery efficiency, organ toxicity, muscle damage, immune response, and the pain. Our data demonstrated the distinct characteristics of the two mRNA delivery platforms, and this knowledge will be indispensable for the further optimization of both systems. 

To date, LNPs have been the most clinically advanced and intensively studied carrier platform for in vivo mRNA delivery. While the mechanism of how LNPs trigger the proinflammatory pathways remains unclear, LNPs have been reported to possess adjuvant properties and contribute to the immune responses [50]. Based on our data, LNPs resulted in remarkably higher levels of protein production, but a greater extent of muscle damage and immune response was also observed. These observations are closely related to a significant exhibition of pain behaviors. Thus, although LNPs displayed desirable characteristics for vaccines, there is still room for further improvement to reduce the excessive immune reponses, which might cause the side effects and the pain.

In contrast, polyplex nanomicelles showed a relatively sustained protein production in the target muscle for up to two weeks. The less invasive manner is supposed to be due to the surface covered by dense PEG palisade, which would minimize nonspecific interactions with the cell membrane. Although it may not be applicable for mRNA vaccines without any adjuvants, the nanomicelle could be a suitable alternative for mRNA delivery, especially for therapeutic applications in the areas where the inflammation is undesirable.

As a new modality of drug development, mRNA has been chiefly investigated for the vaccines, including those for infectious diseases and cancer vaccines. On the other hand, since mRNA can express any protein, the applications would not be limited to vaccines, but also could widely expand for administering therapeutic proteins [51]. At present; however, the number of the clinical traials for therapeutic mRNA medicines is limited to around 15 cases in the world [52], fewer than that of mRNA vaccines. Roughly half of the trials are for the treatment of solid tumors, and the others are for addressing genetic disorders, which have been considered through gene therapy approaches. Intratumor or intravenous routes with LNPs are used in most cases. Although LNPs should be optimized for each case, the current trials appear to be limited to those in which the risks of toxicity and immunogenicity of the LNPs would be acceptable. Indeed, in the only trial of the mRNA medicine for the regenerative medicine (treatment of myocardial ischemia by VEGF mRNA), the mRNA is administered as naked mRNA without carriers [53]. It is apparent that the inflammation at the injected site should be strictly avoided for inducing tissue regeneration. In this regard, the LNPs are disadvantaged.

Thus, it is important to have options for mRNA delivery. The nanomicelle presented in this study is a less risky alternative. For the future, the delivery method would be individually determined for each medical purpose. In this study, we used representative conditions to form the carriers: an N:P ratio of 8:1 was used for nanomicelles based on our previous studies where the nanomicelles were administered to various organs or tissues [12,33]. For LNP, we followed the protocols provided by Moderna, which used an N:P ratio of 5.67:1 [31]. As shown in Appendix A, they both had the size of less than 100 nm, with almost neutral surface charge. Thus, the differences between the two carriers presented in this study would not be attributed to the difference in the physicochemical properties. It should be noted that, in this study, we used a commercially available FLuc mRNA with complete uridine depletion by 5moU (5-methoxyuridine), as depletion by 5 moU has been reported to result in higher mRNA stability and reduced immune responses among all nucleoside modifications [54]. Although modified mRNA may be advantageous in terms of increased stability, unmodified mRNA with intrinsic immunogenicity may help induce effective immunity. To complicate things even further, the effects of nucleoside modification of mRNA may be influenced by the chemical profile of ionizable lipids in the LNPs, as recently reported [55]. In addition, in our previous studies with nanomicelles aiming at disease treatment, we occasionally used unmodified mRNAs, further highlighting the less immunogenic manner of the nanomicelles. Altogether, we believe that the design of the mRNA should be as critical as the delivery platforms for the development of mRNA-based therapies.

In conclusion, by demonstrating the distinct characteristics of two mRNA delivery platforms LNP and polyplex nanomicelle, this study implies the diverse possibilities offered by them, and supports the idea that the ongoing optimization of mRNA delivery systems should be tailored to meet specific clinical needs.

## Figures and Tables

**Figure 1 pharmaceutics-15-02291-f001:**
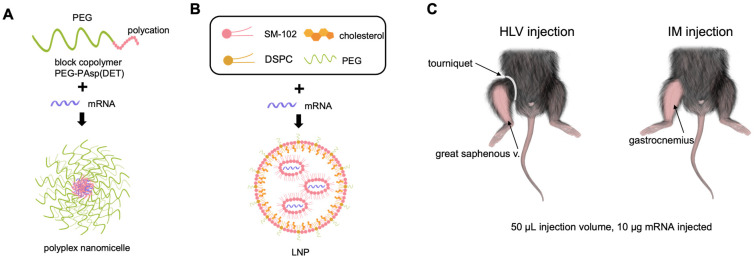
Schematic illustration of FLuc mRNA-loaded polyplex nanomicelle (**A**) and LNP (**B**) when administered by HLV or IM injection (**C**).

**Figure 2 pharmaceutics-15-02291-f002:**
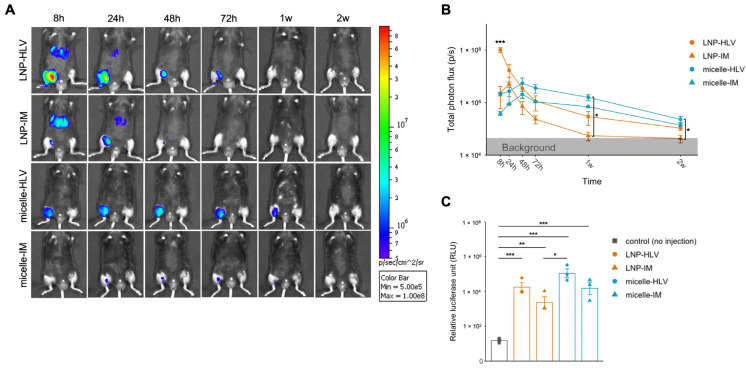
Luciferase expression profile after IM or HLV administration of mRNA delivered by LNP or nanomicelle. (**A**) Representative IVIS images of luciferase expression. Color spectrum represents the intensity of the bioluminescence. (**B**) Time course of luciferase expression observed at 8 h, 24 h, 48 h, one week, and two weeks after administration. (**C**) Luciferase assay performed on muscles at the end of the IVIS experiments. Data are presented as mean ± SEM, *n* = 3. *** *p* < 0.001, ** *p* < 0.01, * *p* < 0.05. One-way ANOVA followed by Tukey’s honest significance test.

**Figure 3 pharmaceutics-15-02291-f003:**
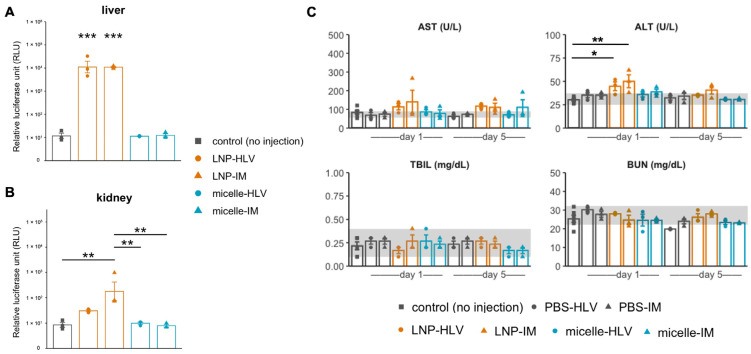
Evaluation of organ toxicity by luciferase assay and blood biochemistry. Luciferase assay was performed on the livers (**A**) and kidneys (**B**) collected 24 h after administration (*n* = 3). (**C**) Blood biochemistry tests on blood samples collected 1 and 5 days after administration (*n* = 3 and *n* = 6 for the control group). Grey bands represent the referential normal range indicated by the animal supplier (Japan SLC Inc.) and previous reports. Data are presented as mean ± SEM. *** *p* < 0.001, ** *p* < 0.01, * *p* < 0.05. One-way ANOVA followed by Tukey’s honest significance test.

**Figure 4 pharmaceutics-15-02291-f004:**
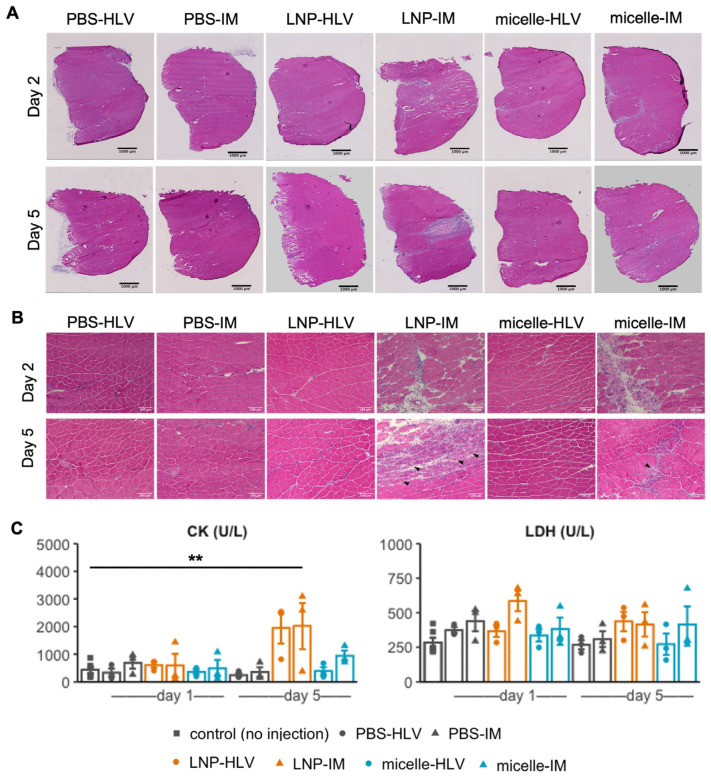
Histological changes in the muscle 2 and 5 days after administration of LNP or nanomicelle at 4× (**A**) and 20× (**B**) magnifications. Arrowheads indicate centrally nucleated fibers. (**C**) Levels of CK and LDH in the blood samples collected 1 and 5 days after administration (*n* = 3 and *n* = 6 for the control group). Data are presented as mean ± SEM. ** *p* < 0.01, one-way ANOVA followed by Tukey’s honest significance test.

**Figure 5 pharmaceutics-15-02291-f005:**
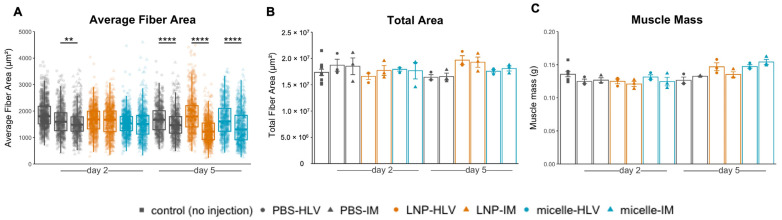
Average fiber area (**A**), total fiber area (**B**), and muscle mass of the muscles collected at 2 and 5 days after administration (*n* = 3 and *n* = 6 for the control group). For (**A**), 200 to 250 fibers from each mouse were analyzed. Data are presented as mean ± SEM. **** *p* < 0.0001, ** *p* < 0.01. Student’s *t*-test was performed for (**A**), and one-way ANOVA followed by Tukey’s honest significance test was performed for (**B**,**C**).

**Figure 6 pharmaceutics-15-02291-f006:**
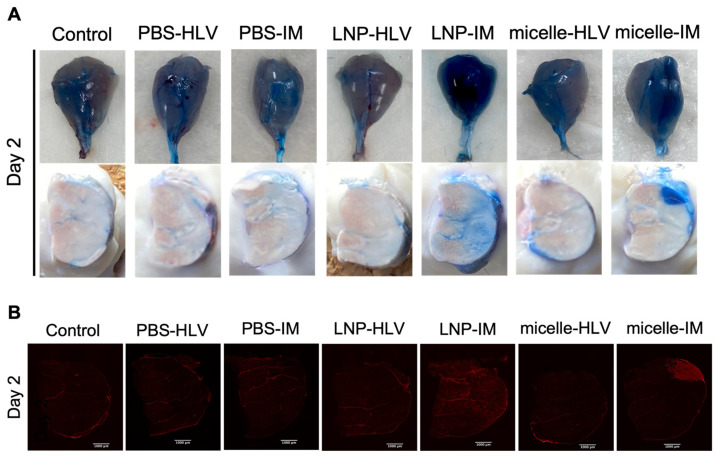
Representative images of EBD staining in the muscle tissue ((**A**), upper panel), muscle transverse section ((**A**), lower panel), and muscle cross-section under a fluorescence microscope (**B**). Samples were collected 2 days after HLV or IM administration of Fluc mRNA-loaded LNP or nanomicelle.

**Figure 7 pharmaceutics-15-02291-f007:**
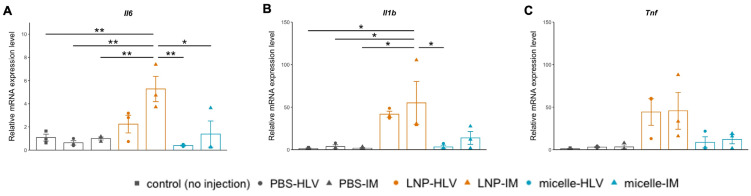
Relative mRNA expression of inflammatory cytokines IL-6 (**A**), IL-1β (**B**), and TNF-α (**C**) 24 h post administration. Results were normalized to the expression levels of β-actin. Data are presented as mean ± SEM, *n* = 3. ** *p* < 0.01, * *p* < 0.05. One-way ANOVA followed by Tukey’s honest significance test.

**Figure 8 pharmaceutics-15-02291-f008:**
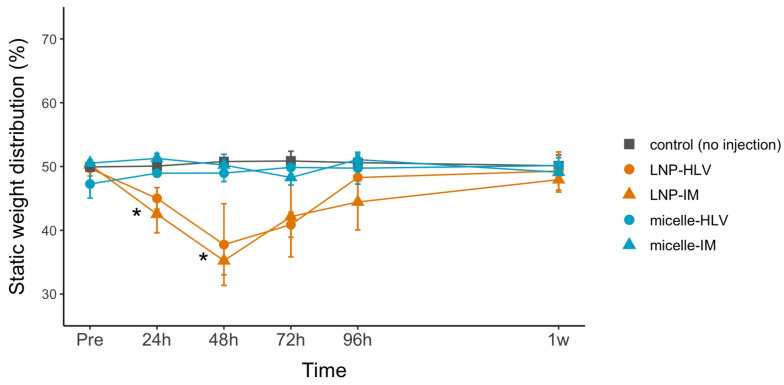
Pain behavior evaluated by the incapacitance test. The distinct difference in weight distribution between the injected and noninjected hindlimb can be interpreted as the animal’s adaptation to pain. Data are presented as mean ± SEM, *n* = 3. * *p* < 0.05, one-way ANOVA followed by Tukey’s honest significance test.

**Table 1 pharmaceutics-15-02291-t001:** Primers used for qRT-PCR.

Gene	Forward Primer (5′ to 3′)	Reverse Primer (5′ to 3′)
*Il6*	TACCACTTCACAAGTCGGAGGC	CTGCAAGTGCATCATCGTTGTTC
*Il1b*	TGGACCTTCCAGGATGAGGACA	GTTCATCTCGGAGCCTGTAGTG
*Tnf*	CCACGTCGTAGCAAACCACC	TTGAGATCCATGCCGTTGGC
*Actb*	GTGACGTTGACATCCGTAAAGA	GCCGGACTCATCGTACTCC

## Data Availability

The data presented in this study are available on request.

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
