# Peer review of "Comprehensive Evaluation of Lipid Nanoparticles and Polyplex Nanomicelles for Muscle-Targeted mRNA Delivery"

_pharmaceutics, 2023, doi:10.3390/pharmaceutics15092291_

Round 1
Reviewer 1 Report
1. Please clarify the color spectrum located on the left side of Figure 2A in either the caption or the main text.
2. The positively charged lipid shell or PEG-PAsp(DET) polymer could potentially bind with negatively charged cell membranes or biomacromolecules through nonspecific electrostatic interactions, causing nonspecific bioeffects. Did the authors test empty LNP or polyplex nanomicelle without Fluc mRNA using intramuscular (IM) or hydrodynamic limb vein (HLV) injection method? It would be beneficial if the authors could address these aspects in the manuscript.
3. Figures
1) In Figure 1, the caption was not correct. Figure 1A should be polyplex nanomicelle and Figure 1B should be LNP.
2) A few figures include background frames, borders, and unnecessary grids, which impact the overall clarity and presentation of the visuals. Could the authors adjust them for better clarity?
The study was well-executed and the manuscript is well-written.
Reviewer 2 Report
The study is about the encapsulation of mRNA sequences into either Lipid nanoparticles (LNPs) or nanomicelles (PEG-PAsp(DET)). The manuscript is outstanding mainly in vitro and in vivo experiments. The data is presented in a fluent way. 2 different carriers were compared in terms of their delivery efficiency, organ toxicity, muscle damage, and immune reaction. The following could be expressed more clearly:
-The characterization of prepared NPs is not explained or demonstrated clearly. Even if they were formulized according to previous procedures, the size, surface charge, or morphology of NPs should be given.
-In the discussion part, comparing the predefined parameters and properties of NPs with the literature would be better.
-It needs to be explained why the experiments were continued until day 5.
-How did the authors measure and determine the total amount of mRNA encapsulated into NPs and did they use these 2 different NPs in equivalent conditions?
Moderate editing of English language required
Reviewer 3 Report
Overall this manuscript is well written. And the experiments are designed logically. I only have 2 minor suggestions.
1. The author used the Moderna LNPs recipe, but they change the PEG to GM-020 instead of using DMG-PEG-2000. Is there any reason why the author chose this PEG?
2. What about the weight change of the mice? The author should provide this data to indicate the LNPs or nanomicelles are safe to use.
English is fine to understand
